# Effects of Different Preservation Techniques on Microbial and Physicochemical Quality Characteristics of Sauced Beef Under Chilled Storage

**DOI:** 10.3390/foods14071175

**Published:** 2025-03-27

**Authors:** Yiling Pan, Xiangnan Xue, Ying Wang, Jinpeng Wang, Wendi Teng, Jinxuan Cao, Yuemei Zhang

**Affiliations:** 1Key Laboratory of Geriatric Nutrition and Health, Beijing Technology and Business University, Ministry of Education, Beijing 100048, China; yilingpan@st.btbu.edu.cn (Y.P.);; 2Beijing Engineering and Technology Research Center of Food Additives, School of Food and Health, Beijing Technology and Business University, Beijing 100048, China

**Keywords:** sauced beef, pepper essential oil, microbial growth, nucleotides catabolism, biogenic amines

## Abstract

This study investigates the effects of different preservation methods—tray packing (control), vacuum packing (T1), and tray packing with 2 mg/mL pepper essential oil (T2)—on the quality of sauced beef during 4 °C storage for 1, 5, 9, and 13 days. The results revealed that T2 significantly inhibited microbial growth, as reflected by reduced total aerobic counts (TACs), minimized lipid oxidation (indicated by lower thiobarbituric acid reactive substances (TBARSs)), and reduced protein degradation (evidenced by decreased total volatile basic nitrogen (TVB-N)). Additionally, T2 delayed the reduction in inosine-5′-monophosphate (IMP) and accumulation of hypoxanthine (Hx), effectively extending shelf life and preserving sensory quality. T1 also showed beneficial effects in limiting oxygen-related spoilage, as demonstrated by lower TAC and TBARS levels. In contrast, the control group showed limited effectiveness in preserving the quality of sauced beef, as indicated by higher microbial counts and more pronounced lipid and protein degradation. These findings provide a theoretical basis for improving sauced beef preservation by highlighting the effectiveness of different packaging methods and the potential of pepper essential oil as a natural preservative.

## 1. Introduction

China, as the world’s top meat producer, ranks third in beef consumption, following pork and poultry [1,2]. The USDA’s Global Beef Market Forecast 2024–2025 report estimates that China’s beef consumption will reach 11.557 million tons by 2025. Sauce-marinated meat products are pre-treated and pre-cooked meats marinated in a seasoned brine with spices and condiments [3]. Due to their vibrant color, appealing aroma, rich taste, and convenience, sauced beef products are highly favored by consumers and are a popular choice among prefabricated meat products. However, sauced beef products are more susceptible to microbial contamination and oxidative spoilage during the storage process, considering their high moisture content and the presence of oxygen, which provide an ideal environment for microbial growth and accelerated fat oxidation [4]. This not only shortens the product shelf life, but also brings economic losses and threatens the health and safety of consumers.

Biogenic amines (BAs) are nitrogenous compounds that are beneficial in moderation, but excessive intake (e.g., histamine > 40 mg) causes headaches, vomiting, blood pressure fluctuations, or even fatal toxicity. Histamine is the most toxic, followed by tyramine. Low-toxicity amines (e.g., cadaverine) may react with nitrites to form carcinogenic nitrosamines [5]. BAs are commonly found in spoiled fish and fermented foods; their toxicity is exacerbated by alcohol, posing higher risks for sensitive individuals (e.g., antidepressant users) [6]. The degradation of nucleotides plays a significant role in evaluating the freshness of meat, as it directly impacts the biochemical changes that occur post-mortem [7]. Nucleotides found in meat primarily originate from the catabolism of ATP, following the degradation pathway adenosine-5′-triphosphate (ATP) → adenosine-5′-diphosphate (ADP) → adenosine-5′-monophosphate (AMP) → inosine-5′-monophosphate (IMP) → inosine (HxR) → hypoxanthine (Hx) → xanthine (Xa) → uric acid (UA). The rate and extent of nucleotide degradation are influenced by factors such as post-mortem handling, storage temperature, and microbial activity. High concentrations of IMP are typically associated with better meat quality, as it contributes to desirable characteristics such as improved flavor and tenderness. However, the further degradation of IMP into HxR and HX is linked to a decrease in meat freshness, as these compounds are associated with a sour or off-flavor [8]. Research conducted by Liu et al. [9] investigated the nucleotide catabolism in both sterile and inoculated forms with four different bacteria carp samples, revealing that autolysis may facilitate the transformation of ATP into IMP, whereas bacteria likely play a significant role in the hydrolysis of IMP. Therefore, the shelf life and sensory quality of sauced beef can be assessed by monitoring nucleotide degradation products and biogenic amine formation.

Currently, commonly used preservation methods for sauce-marinated meat products include the addition of preservatives, packaging technology (vacuum packaging or air-conditioned packaging), technology (heat, microwave, irradiation, high pressure processing or cold atmospheric plasma), and low-temperature preservation [4]. Huang et al. [10] used gamma irradiation to treat smoked chicken breast and demonstrated that irradiation doses >3 kGy were effective for sterilization and reducing protein oxidation. However, they also accelerate lipid oxidation. Chen et al. [11] found that a 3 or 5 h papaya extraction and 450 MPa high-pressure processing caused more than a 6 log *Salmonella* reduction. Cold atmospheric plasma sterilization technology is an emerging non-thermal processing technology that utilizes the surrounding media to generate light, ions, and free radicals to carry out strong sterilization [12]. However, a high temperature can degrade the nutritional value and flavor of meat, and non-thermal technologies like irradiation, high-pressure processing, and cold atmospheric plasma require high-end equipment. Therefore, refrigerated storage, along with tray and vacuum packaging, remains widely used in the meat industry due to its cost-effectiveness and convenience.

Vacuum packaging has been shown to significantly extend the shelf life of sauce-marinated meat products by removing oxygen, thereby inhibiting the growth of aerobic microorganisms. Studies have shown that vacuum packaging slows down fat oxidation and protein degradation, preserving the color and flavor of the products [13,14]. Researchers found that low-temperature storage reduces BAs contents in meat, with vacuum packing at 3 °C extending shelf life by 4–5 days compared to non-vacuum packaging [15]. Recent research has further optimized this technology by investigating the effects of different vacuum levels, packaging materials, and storage temperatures on preservation efficacy [16,17,18]. Meanwhile, as the consumer awareness of food safety continues to improve, plant essential oils have become a research hotspot as natural antimicrobial and antioxidant agents for preserving sauce-marinated meat products. Essential oils from plants, such as cinnamon, clove, and thyme, have been demonstrated to effectively inhibit common foodborne pathogens and spoilage microorganisms [19,20,21]. Moreover, the combined application of multiple essential oils has been found to create synergistic effects, further enhancing preservation outcomes [22]. Additionally, advancements in nanoemulsion and microencapsulation technologies have improved the stability and controlled release of essential oils, making them more effective in practical applications [23,24]. Wei et al. [25] prepared a Pickering emulsion of ZnO nanoparticles and star anise essential oil and combined it with radiofrequency treatment for vacuum-packed chicken feet, which reduced the oxidation of lipids by 20% and proteins by 10%, respectively. Current research on sauced beef focuses on the effects of different marinating recipes and processing techniques on quality characteristics and consumer acceptance [26,27,28]. However, to date, there has been limited research on preservation methods that influence the physicochemical quality development of sauced beef during chilled storage.

This study aimed to investigate the effects of different preservation methods, tray packing, vacuum packing, and tray packing with added pepper essential oil (PEO), on the microbial growth, protein degradation, lipid oxidation, BA accumulation, and nucleotide catabolism of sauced beef stored at 4 °C. The main indicators included TAC, pH, TBARS, TVB-N, BAs, nucleotides, and organoleptic qualities, in order to provide a theoretical basis for preventing and controlling the spoilage of sauced beef and extending its shelf life in future applications.

## 2. Materials and Methods

### 2.1. Sample Preparation

In this study, sauced beef was used as the test material. Beef and condiments such as ginger, bay leaf, aniseed, Chinese prickly ash, and cinnamon were purchased from Xinfadi market in Fengtai District, Beijing, China. Pepper essential oil (PEO) was purchased from Huiwen Spice Co. in Qingyuan District (Ji’an, China). Tenderloins from cattle with an average age of 1.5 to 2 years were used as raw meat. Fresh beef was covered with crushed ice and transferred to the laboratory within 1 h. As shown in Figure 1, the beef was processed into sauced beef following a standardized procedure: the surface fascia was removed, the meat beef was cut into uniform cubes (2.5 cm × 2.5 cm × 2.5 cm), and the cubes were blanched for 5 min and washed to remove foam; then, the beef was marinated for 1 h with a spice pack containing 1% ginger, 0.1% Chinese prickly ash and aniseed, 0.05% bay leaf and cinnamon, 1% salt and sugar, and 0.5% soy sauce, calculated based on the weight of the raw meat.

All sauced beef was mixed well after preparation and randomly divided into three groups for packaging. Samples in the control group were packed in polypropylene (PP) trays and sealed with polyethylene (PE) cling film; samples in treatment group 1 (T1) were vacuum-packed in bags made of PE; and samples in treatment group 2 (T2) were packed in PP trays and sealed with PE cling film after spraying the surface of the meat samples with 2 mg/mL PEO. The PE cling film has a thickness of 25 μm, an O_2_ transmission rate (OTR) of 4000 cm^3^/ (m^2^·day·atm), and a water vapor transmission rate (WVTR) of 15 g/m^2^·day. In comparison, the PE vacuum bag has a thickness of 80 μm, an OTR of 50 cm^3^/ (m^2^·day·atm), a WVTR of 2 g/m^2^·day, and a vacuum pressure of approximately 0.77 atm. All samples were stored in a biochemical incubator at 4 °C and 40% relative humidity. Five trays of meat were randomly selected from each group on the 1st, 5th, 9th, and 13th day. Samples for sensory evaluation and the detection of microbial counts and pH value were determined on the same day, and the remaining samples were cut with sterile surgical scissors and packed in tinfoil for freezing at −80 °C for other physicochemical analyses.

### 2.2. Sensory Evaluation

The sensory evaluation of sauced beef was conducted by 10 sensory evaluators (5 males and 5 females, aged 20–28), following the method of Zheng et al. [28] with minor modifications. Odor descriptions included sauciness, meatiness, spiciness, putrefactiveness, sourness, and rancid odor of the beef. Taste descriptions included umami, sourness, texture, residual taste, and overall acceptability of the beef. Evaluators were asked to rate sensory indicators on a scale of 1–10 (1 means not perceptible, 10 means maximum sensory intensity). If the overall acceptability score is below 5, it is considered corrupt. The evaluators did not eat or smoke for 2 h before the test, and the evaluators performed the evaluations separately. Prior to sensory evaluation, the samples were removed from the 4 °C storage condition and allowed to equilibrate at room temperature for 1 h. Then, the meat was cut into slices that were 2 mm thick for sensory evaluation, and participants rinsed their mouths with water between tasting each sample. Sensory evaluation was approved by the Ethics Committee of Beijing Technology and Business University.

### 2.3. Determination of TAC, pH, TBARS and TVB-N Contents

Samples were homogenized (Ultra Turrax T25, IKA, Staufen, Germany) for 30 s by adding a 10-fold sterile NaCl solution, followed by serial dilution (1:10, NaCl solution). Samples of the appropriate dilution (0.1 mL) were selected and spread on plate count agar and incubated for 72 ± 2 h at 30 ± 1 °C. TAC was expressed as log CFU/g. The pH values of the collected samples were measured using a pH meter (Mettler Toledo Instruments Co., Ltd., Shanghai, China). TBARSs were quantified to determine the degree of lipid oxidation, as described by Cheng et al. [29]. Results are expressed as mg malonaldehyde (MDA)/kg sample. TVB-N was determined using a fully automated Kjeldahl nitrogen detector (Hanon K9840, Jinan, China) followed the method of Liu et al. [9].

### 2.4. Determination of BAs

Putrescine (PUT), cadaverine (CAD), and histamine (HIS) were identified and quantified using standards from Sigma-Aldrich Trading Co., Ltd. (Shanghai, China). The preparation of BAs from beef samples followed the method of Liu et al. [9] with minor modifications. Briefly, 5 g of sauced beef was mixed with 10 mL of HClO_4_ (0.5 M) and then homogenized at 12,000 rpm for 12 s, followed by centrifugation at 8000× *g* for 5 min. This process was repeated once, and all supernatants were combined and adjusted to a volume of 25 mL. Extracts were stored at −18 ± 1 °C until analysis. After pre-column derivatization with dansyl chloride, the extracts were identified and quantified by high-performance liquid chromatography (HPLC) (Agilent 1260, Agilent Technologies, Inc., Santa Clara, CA, USA) equipped with a COSMOSIL 5C18-PAQ column (4.6ID 250 mm), as reported by Fan et al. [30].

### 2.5. Nucleotide-Related Compounds Determination

ATP, ADP, AMP, IMP, HxR, Hx, and Xa were extracted according to Liu et al. [31] with minor modifications. The extraction procedure for nucleotide-related products was as follows. Briefly, 2.5 g of ground beef was mixed with 10 mL of 0.5 M perchloric acid solution and the sample container was placed in crushed ice and homogenized at 12,000 rpm for 18 s, followed by centrifugation at 4 °C, 5000× *g* for 5 min. After the removal of the supernatant, the precipitate was washed with 5 mL of 0.5 M perchloric acid and centrifuged (5000× *g*, 5 min, 4 °C) and the washing process was repeated twice. All supernatants were combined, neutralized to pH 6.35–6.45 with 5 M KOH solution, and centrifuged at 5000× *g* for 5 min. The supernatant was collected and concentrated to 25 mL with perchloric acid (pH 6.40). The solution was filtered through a 0.22 μm aqueous filter and analyzed by HPLC. A COSMOSIL 5C18-PAQ column (4.6ID 250 mm) with phosphate buffer (0.05 M, pH 6.8) was used at a flow rate of 1 mL/min. The eluent was monitored at 254 nm and compared with the commercially obtained standard (Sigma-Aldrich Trading Co., Ltd., Shanghai, China).

### 2.6. Statistical Analysis

All results were carried out in three replicates. The statistical analysis was conducted via a one-way analysis of variance (ANOVA) using SPSS software (v.19; SPSS Inc., Armonk, NY, USA). Differences between means were assessed by Duncan’s multiple range test at the significance level of 5%. Origin 2025 software (OriginLab Co., Northampton, MA, USA) was used to plot the data.

## 3. Results and Discussion

### 3.1. Sensory Evaluation

The sensory evaluation scores of the sauced beef stored at 4 °C for 13 days are shown in Figure 2. In Figure 2a–c, which focus on odor, for the control, the intensity of the rancid odor increased significantly over the storage period, while the sauce and spice odors decreased, indicating a reduction in the overall sensory quality of the sauced beef over time. T1 also showed an increase in rancid odor, but at a slower rate compared to the control, with more stable sauce and spice odors. T2 maintained relatively lower levels of a sour odor and rancid odor throughout the storage period. The sauce odor decreased from 8.7, 8.8, and 8.7 points on day 1, to 1.5, 4.8, and 1.9 points on day 13 in the control, T1, and T2 groups, respectively. The rancid odor increased from not perceptible on day 1 to 9.4, 3.9, and 2.6 points on day 13. Looking at Figure 2d–f, related to taste and texture, the control experienced a decline in umami flavor and overall acceptability as storage progressed; the residual taste also decreased, and the sourness increased. In terms of texture, the control showed a negative change over time. T1 had the highest umami flavor and overall acceptability scores among the three groups during most of the storage period. The texture and residual taste for T1 samples were also more stable, with less of an increase in sourness. For T2, there were relatively minor changes in umami flavor and residual taste compared to control, but still a gradual decrease in overall acceptability was observed. Overall acceptability decreased from 8.9, 9.1, and 9.0 points on day 1 to 4.4, 6.9, and 5.9 points on day 9 for the control, T1, and T2 groups, respectively. Furthermore, overall acceptability declined to 3.5 and 2.2 points on day 13 for the T1 and T2 groups, respectively. It can be inferred that oxygen exposure in tray packing might accelerate oxidation and microbial proliferation, leading to undesirable organoleptic changes [32]. In contrast, T1 and T2 showed different patterns due to their distinct packaging characteristics, e.g., the low-oxygen environment in T1 was effective in preserving the beef aroma and taste, and the addition of 2 mg/mL PEO in T2 effectively inhibited the formation of rancid odors, likely due to its antibacterial and antioxidant properties [33]. The stable sensory characteristics of T2 suggest that PEO can be a useful additive for preserving the sensory quality and extending the shelf life of sauced beef. Sensory quality is a primary driver of consumer choices, and these results highlight the potential of improved packaging methods to enhance the overall eating experience of meat products, aligning with previous studies that emphasized the importance of sensory attributes in food [34].

### 3.2. TAC, pH, TBARS, and TVB-N Contents

As shown in Figure 3a, the TAC values for all three groups increased over the storage period. The control group had the highest TAC values throughout the storage time, reaching 4.93 log CFU/g by day 9, nearing the national standard limit of 5.00 log CFU/g, suggesting a shelf life not exceeding 9 days. T1 had a slower increase in TAC compared to the control, and then entered into a rapid growth on day 5, reaching 3.69 log CFU/g at day 9 and 6.29 log CFU/g at day 13. T2 had significant antibacterial effects, reaching only 3.48 log CFU/g at day 13.

From Figure 3b, the pH values of the T1 and T2 groups showed a slight initial decrease, followed by a significant increase over time (*p* < 0.05). For the control group, the pH value was very stable during the pre-storage period and increased significantly on day 13 (*p* < 0.05). There were no significant differences among the groups at the same time (*p* > 0.05). However, as storage progressed, the three groups had the highest pH values on day 13. The pH value of the samples increased from 6.35, 6.41, and 6.37 on day 1 to 6.55, 6.46, and 6.50 on day 13 in the control, T1, and T2 groups, respectively. The change in pH values of sauced beef may be due to the fact that lactic acid produced by microorganisms such as lactobacilli, which break down sugars, causes a slight decrease in pH values at the beginning of storage [35]. Carbonyl compounds from protein oxidation can further oxidize to form acids [36]. As storage time increases, alkaline substances produced by protein breakdown and microbial metabolites lead to an increase in pH values [37].

TBARS values have been widely employed to assess the levels of secondary lipid oxidation products, primarily concentrating on MDA, which is a crucial byproduct arising from the breakdown of primary lipid oxidation processes that take place in diverse biological and food systems [38]. According to Figure 3c, the TBARS values of the control and T2 groups in the pre- and mid-storage periods (day 1–day 9) showed a significant increasing trend (*p* < 0.05) with the prolongation of storage time. However, the TBARS values of the late storage period (day 9–day 13) were no longer significantly elevated (*p* > 0.05). TBARS values in control and T2 groups increased by 108% and 88%, respectively, on day 13 compared to day 1, reaching 3.0 mg MDA/kg and 2.7 mg MDA/kg, respectively, on day 13. A TBARS value ≥ 2.0 mg MDA/kg indicates that the meat is not only significantly rancid but also deemed unfit for consumption [39]. The TBARS values of the T1 group were very stable (*p* > 0.05) throughout the storage period, reaching a maximum of 1.4 mg MDA/kg on day 13. This suggests that vacuum packaging can effectively prevent lipid oxidation, and the increase in TBARS values in sauced beef is mainly caused by oxygen exposure, while microbial activity has little effect. These results are also in agreement with Jaberi et al. [40], who reported that after the application of vacuum and high-oxygen modified atmosphere packaging (80% O_2_ + 2% CO_2_) on minced water buffalo meat at 2 °C for 14 days, TBARS values in vacuum-packaged samples were below 0.2 mg MDA/kg and over 6.0 mg MDA/kg in modified atmosphere packaging samples.

TVB-N is commonly used as a biomarker for protein and amine degradation, as well as a marker of spoilage in meat products [41]. The endogenous enzymes were largely inactivated after the beef was sauced by heating, and the increase in the TVB-N content was mainly caused by the growth and multiplication of microorganisms. As shown in Figure 3d, the TVB-N content increased from an initial value of 8.40 mg/100 g to 14.28, 13.35, and 10.45 mg/100 g on day 13 in the control, T1, and T2 groups, respectively. TVB-N values increased rapidly in the control and T1 groups but were significantly slower in the T2 group (*p* < 0.05). These findings are consistent with the results of TAC changes, indicating that the use of PEO can effectively inhibit microbial growth and thus prolong meat quality. Recent research on the application of black pepper essential oil in meat preservation has revealed its active components, such as piperine and various terpenes, which contribute to its efficacy in extending shelf life [42,43]. The mechanisms underlying its preservative effects include the disruption of microbial cell membranes and the scavenging of free radicals, which can delay oxidative rancidity [33,44].

Packaging techniques significantly affected the sauced beef quality. Tray packaging increased oxygen exposure, promoting aerobic microorganisms and higher TAC values. The increased microbial activity also contributed to higher pH changes and greater protein and lipid degradation, as indicated by the higher TVB-N and TBARS levels. Vacuum packaging reduced oxygen exposure, leading to the lowest TBARS values throughout the storage period, indicating the better preservation of lipid stability. The addition of 2 mg/mL of PEO in T2 effectively inhibited microbial growth, resulting in the lowest TAC and TVB-N values at the end of storage. The results of TAC, TVB-N, and TBARS analyses were consistent with the trends in the sensory scores, indicating that the sauced beef shelf life in the control group was under 9 days, while the shelf life in the T1 and T2 groups was under 13 days. This suggests that PEO’s antimicrobial properties can play a crucial role in extending the shelf life of sauced beef by controlling microbial activity and reducing spoilage.

### 3.3. Changes in BAs

BAs are a class of nitrogen-containing organic compounds that are produced mainly from free amino acids by decarboxylation reactions under the action of microorganisms [45]. Two types of BAs, PUT and CAD, are usually found at high levels in spoiled meat. HIS is one of the most toxic biogenic amines, and its excessive ingestion can cause allergic reactions, headaches, nausea, and breathing difficulties [46]. Changes in BAs in these three groups are shown in Figure 4, in which only PUT and HIS were detected. The initial concentrations of PUT and HIS were very low in all groups; this may be due to the fact that the cooking temperatures resulted in protein coagulation and denaturation, as well as a reduction in the free amino acid content, which in turn reduced the production of biogenic amines. For the PUT value, as shown in Figure 4a, on day 1, there were no significant differences (*p* > 0.05) among the three groups, with values all around 0.2–0.3 mg/kg. As the storage time increased to day 13, the control and T1 groups showed a significant increase compared to T2 (*p* < 0.05). The control group on day 13 had the highest PUT value, reaching 0.66 mg/kg, and T1 reached 0.48 mg/kg, indicating rapid PUT production in the control and T1 groups. The rapid accumulation of PUT in the control and T1 groups could be linked to the higher levels of microbial growth, especially in the tray and vacuum packaging groups, which allowed more microbial proliferation due to differences in oxygen availability. In contrast, the relatively stable PUT levels in T2 suggest that the antimicrobial effects of PEO might have inhibited microbial growth, thus preventing the accumulation of PUT. Regarding the HIS values, as shown in Figure 4b, there was an overall increasing trend in the HIS content in all groups with increasing storage time, but there was no significant difference among the three groups during storage (*p* > 0.05). Fluctuations in the levels of BAs can be elucidated by microflora exhibiting diverse biochemical capabilities for metabolizing amino acids. Numerous studies have demonstrated that various *Enterobacteriaceae*, *Pseudomonas* spp., and certain species of *Lactobacillus*, *Enterococcus*, and *Staphylococcus* are especially implicated in the synthesis of Bas [32]. Notably, *Enterobacteriaceae* possess elevated lysine and ornithine decarboxylase activities, resulting in the production of substantial quantities of CAD and PUT [47]. China currently only limits HIS in fresh and frozen aquatic animal products to less than 200 mg/kg. Reference can be made to EU recommendations (total BAs ≤ 1000 mg/kg) or specific BAs (e.g., HIS ≤ 100 mg/kg). Our results were well below the standard limits.

### 3.4. Nucleotide-Related Compounds

As shown in Figure 5a–c, the concentrations of ATP, ADP, and AMP in the three groups showed a stable trend during the storage period, with no significant differences between and within groups (*p* > 0.05). This was different from the raw meat, where a sharp decline in these metabolites was reported, likely due to the degradation of ATP, ADP, and AMP during the post-slaughter rigor mortis period [7]. The cooking process could deactivate the endogenous muscle enzymes (such as ATPase and AMP deaminase), resulting in minimal residual activity of these enzymes in the final meat products. Thus, the contents of ATP, ADP, and AMP in sauced beef showed almost no change during chilled storage.

As shown in Figure 5d, the IMP contents of the three groups ranged from 27.86 to 28.18 mg/100 g on the first day of storage. By the 5th day, the content increased slightly, between 29.36 and 30.16 mg/100 g, for three groups. However, as storage progressed, a rapid decline in IMP concentration was observed, with values of 20.20, 23.52, and 26.48 mg/100 g (*p* < 0.05) by the 13th day in control, T1, and T2 groups, respectively. The initial slight rise in IMP content may be attributed to residual AMP deaminase activity in beef, which supports the conversion of AMP into IMP, thereby facilitating its accumulation. Additionally, inadequate sterilization or contamination in the storage environment could allow certain bacterial strains to proliferate, which are capable of excreting purines and directly generating IMP through microbial fermentation [48]. The 5′-nucleotidase secreted by microorganisms at the late stage of storage catalyzed the degradation of IMP to HxR, leading to a decrease in IMP content. However, higher contents of IMP were found in the T1 and T2 groups compared to the control group, indicating a reduced rate of IMP degradation during storage. This is consistent with the changes in TAC and TVB-N. As shown in Figure 5e, the initial HxR content of sauced beef was relatively low, ranging from 15.43 to 15.98 mg/100 g. During the storage period, the HxR levels in the control and T1 groups increased, reaching a plateau on the 9th and 13th day, with the highest concentrations of 18.65 and 17.82 mg/100 g, respectively. The highest concentration in the T2 group was only 15.83 mg/100 g. HxR was degraded into Hx by the action of purine nucleoside phosphorylases and inosine nucleotidases, with Hx serving as a critical indicator of meat product spoilage. In Figure 5f, the initial content of Hx in the three groups was recorded at 10.20 to 10.40 mg/100 g. The content of the control group increased significantly after 9 days of storage, peaking at 12.51 mg/100 g on the 13th day. The Hx levels in the T1 group showed a slight upward trend over time, although there was no significant difference between the groups in both T1 and T2 during the storage period (*p* > 0.05). The continued degradation of Hx, facilitated by xanthine oxidase, was also evident. As illustrated in Figure 5g, Hx was further converted to Xa by xanthine oxidase, with Xa levels positively correlated with storage time, culminating in a maximum concentration of 2.19 mg/100 g in the control group on day 13, followed by the T1 and T2 groups at 2.04 and 1.94 mg/100 g, respectively. This suggests that control group experienced a more rapid degradation of IMP, leading to higher Hx and Xa accumulation, which is typically associated with more advanced spoilage. In contrast, the relatively stable Hx and Xa levels in the T1 and T2 groups indicate that these treatments helped to slow down the spoilage process, potentially by limiting microbial activity or minimizing nucleotide degradation. Similarly, Komatsu et al. [7] discovered a negative correlation between IMP levels and aging time in post-slaughtered beef, characterized by an initial increase in HxR content, followed by a decrease, while Hx levels continued to rise. Fan et al. [30] also found a similar changing trend of nucleotide-related compounds in fish fillets when stored at 4 °C.

### 3.5. Relationship Between Storage Time and TAC, pH, TBARS, TVB-N, BAs, and Nucleotide-Related Compounds of Sauced Beef

In order to analyze the role of microbial counts in the formation of physicochemical indicators during storage, the correlation between microbial counts and physicochemical quality characteristics were calculated using the Pearson correlation coefficient. As shown in Figure 6, TAC has a significant correlation with various physicochemical indicators in sauced beef. For instance, it has a strong positive correlation with storage time, TVB-N, PUT, HIS, HxR, Hx, and Xa, indicated by large red circles with ** or * symbols, suggesting that the contents of these indicators increased as the storage time extended. In contrast, TAC shows a negative correlation with ATP, ADP, AMP, and IMP, shown by blue circles, suggesting a decrease in their contents over time. Among the physicochemical indicators, there are also notable correlations. TVB-N in the sauced beef has a positive correlation with several indicators like PUT, HIS, and Hx. Some indicators, such as pH, show relatively weak correlations with other parameters in certain cases. The positive correlations between storage time and indicators like TAC, TVB-N, PUT, and HIS imply that with the extension of storage time, the microbial growth and spoilage processes in sauced beef are enhanced. Over time, microbial proliferation leads to an increase in metabolites associated with spoilage, which is consistent with the general understanding of meat deterioration during storage. The correlations among different physicochemical indicators provide insights into the complex biochemical processes occurring in sauced beef. For example, the close relationship between TVB-N and other spoilage-related indicators indicates that they can be used together to monitor the spoilage status of sauced beef. Understanding these correlations can help in establishing more comprehensive quality control systems and predicting the shelf life of sauced beef more accurately.

## 4. Conclusions

This study compared the effects of different preservation methods on the quality of sauced beef, highlighting the key factors influencing its shelf life and quality. The degradation of nucleotides and proteins and the BA formation in sauced beef were primarily driven by microbial growth, while lipid degradation was mainly influenced by oxygen exposure. The results showed that 2 mg/mL of PEO could significantly inhibit microbial growth, reduce lipid oxidation and protein degradation, and delay IMP degradation and Hx accumulation, ultimately extending the shelf life of sauced beef. Vacuum packaging also had positive effects in reducing oxygen-related spoilage, as indicated by lower TBARS and TAC levels. Therefore, combining essential oils with vacuum packaging offers a promising method for preserving sauced beef products, enhancing both their shelf life and overall quality, while reducing the need for synthetic preservatives.

## Figures and Tables

**Figure 1 foods-14-01175-f001:**
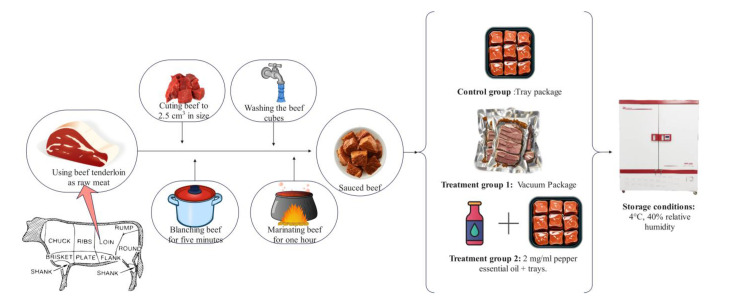
Processing, packaging, and storage of sauced beef samples. (Control: beef packed in trays; T1: beef packed in vacuum; T2: beef packed in 2 mg/mL pepper essential oil + trays).

**Figure 2 foods-14-01175-f002:**
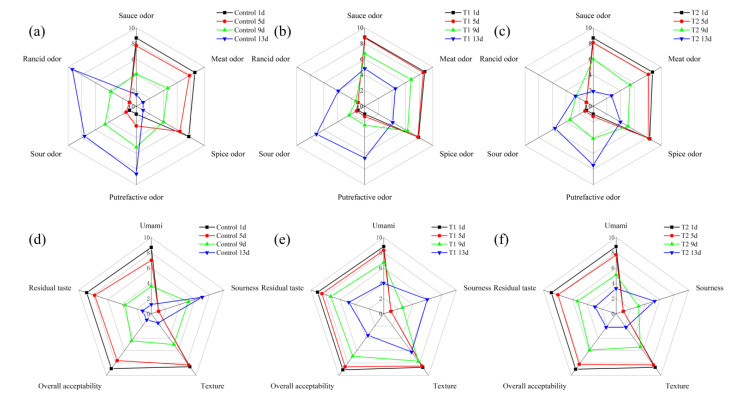
Sensory evaluation radar map of sauced beef storage at 4 °C for 13 days. (**a**–**c**) represent control, T1, and T2 odor attribute scores. (**d**–**f**) represent control, T1, and T2 taste attribute scores, respectively. (Control: beef packed in trays; T1: beef packed in vacuum; T2: beef packed in 2 mg/mL pepper essential oil + trays).

**Figure 3 foods-14-01175-f003:**
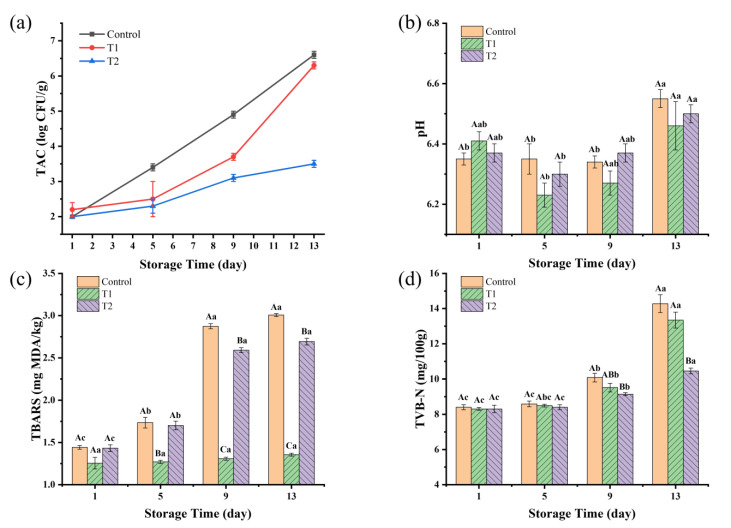
(**a**–**d**) Changes in TAC, pH, TBARS, and TVB-N of sauced beef. Different capital letters indicate significant differences (*p* < 0.05) at the same storage time. Different lowercase letters indicate significant differences (*p* < 0.05) within the same group. (Control: beef packed in trays; T1: beef packed in vacuum; T2: beef packed in 2 mg/mL pepper essential oil + trays).

**Figure 4 foods-14-01175-f004:**
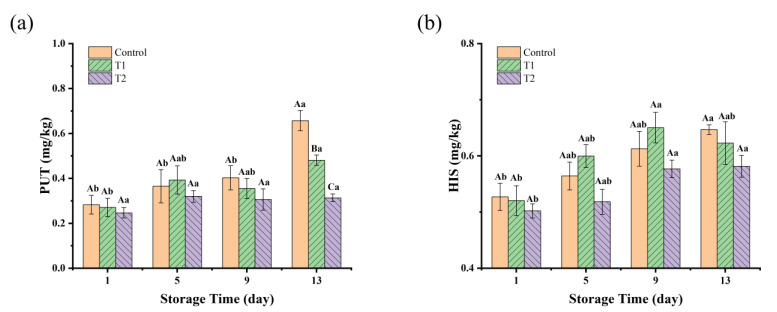
BAs concentration (mg/kg) in sauced beef storage at 4 °C for 13 days. (**a**) Changes in the concentrations of PUT. (**b**) Changes in the concentrations of HIS. Different capital letters indicate significant differences (*p* < 0.05) at the same storage time. Different lowercase letters indicate significant differences (*p* < 0.05) within the same group. (Control: beef packed in trays; T1: beef packed in vacuum; T2: beef packed in 2 mg/mL pepper essential oil + trays).

**Figure 5 foods-14-01175-f005:**
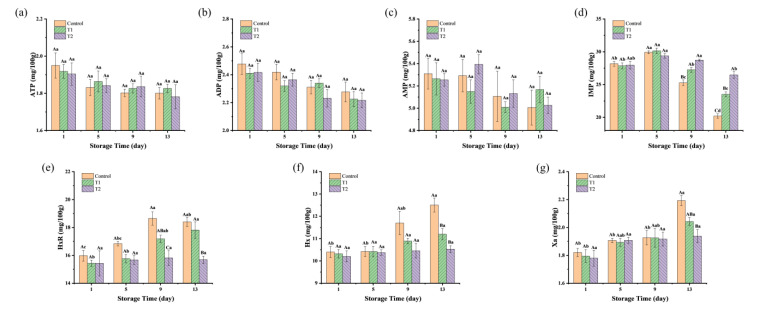
Changes in nucleotide-related compounds and their degradation products in sauced beef stored at 4 °C for 13 days. (**a**–**g**) Changes in ATP, ADP, AMP, IMP, HxR, Hx, and Xa of sauced beef. Different capital letters indicate significant differences (*p* < 0.05) at the same storage time. Different lowercase letters indicate significant differences (*p* < 0.05) within the same group. (Control: beef packed in trays; T1: beef packed in vacuum; T2: beef packed in 2 mg/mL pepper essential oil + trays).

**Figure 6 foods-14-01175-f006:**
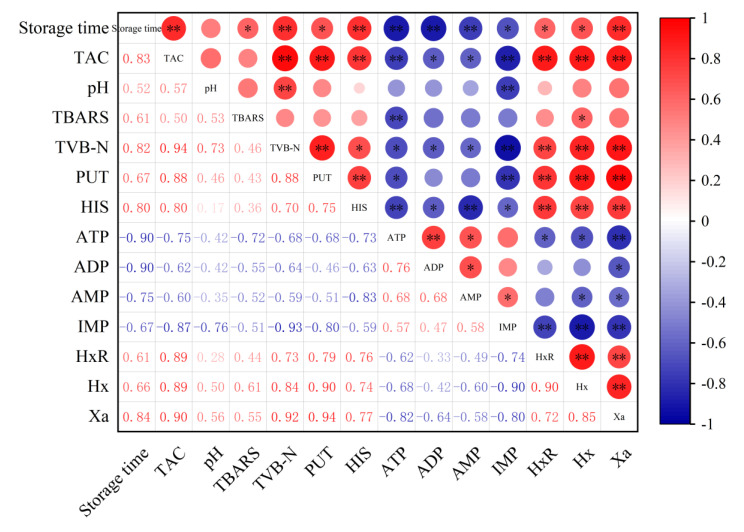
Heatmap of Pearson correlation between physicochemical indicators and storage time in sauced beef. The area of the circle represents the absolute value of the correlation coefficient, where red represents a positive correlation and blue represents a negative correlation. (* *p* ≤ 0.05, ** *p* ≤ 0.01).

## Data Availability

The original contributions presented in this study ar e included in the article. Further inquiries can be directed to the corresponding author.

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
