# Peer review of "Effects of Different Preservation Techniques on Microbial and Physicochemical Quality Characteristics of Sauced Beef Under Chilled Storage"

_foods, 2025, doi:10.3390/foods14071175_

Round 1
Reviewer 1 Report
Comments and Suggestions for Authors
Dear authors,
In this investigation, the effects of different preservation techniques on microbial and physicochemical quality characteristics of sauced beef under chilled storage were evaluated.
This research is marginally novel as essential oils have already been used as preservatives in meat and meat products. However, interesting results can be seen with in-depth discussions provided.
In order for the manuscript to be considered for publication, the following must be reviewed:
Line 17: Provide definitions for IMP and Hx.
Line 49: A connector is needed between “...equipment,” and “refrigerated…”
Line 89: Etc.? Do not use "etc." as it is necessary to mention each of the ingredients so that the research can be reproducible.
Line 94: Brined? With which type of salt?
Lines 110-119: This section is normally placed before the statistical analysis. The sensory analysis lacks information on how it was carried out and whether water and biscuits were supplied between samples.
Lines 117-118: Were the samples served at room temperature? Give more detail on how the sensory analysis was carried out.
Lines 121: Provide more details on how the TAC analysis was carried out.
Line 129: Provide the full name and in parentheses give the acronym.
Line 133: Beef or sauced beef?
Line 134: Repeat? Please describe the methodology in the past tense.
Line 206-207: The authors state: “From Figure 3(b), the pH values of the three groups showed a slight initial decrease followed by a significant increase over time (P < 0.05).” I do not agree, as it can be observed that T1 group remained constant bewteen days 1 to 9.
Line 210: This information cannot be understood. To which group does the value 6.4 belong?
Lines 218-234: Nowhere is it mentioned whether the results obtained consider the beef to be rancid. Please research the oxidation limit that is considered for beef.
Lines 318- 321: Interesting information.
Furthermore, it is essential that the manuscript be reviewed by a native speaker of English.
Comments on the Quality of English LanguageThe English could be improved to more clearly express the research.
Author Response
Line 17: Provide definitions for IMP and Hx.
Response: Thanks for your suggestions. We have revised the abstract to include definitions for "IMP" and "Hx" upon their first mention.
Line 48: A connector is needed between “...equipment,” and “refrigerated…”
Response: Thanks for your suggestions. We have added a connector to improve the flow of the sentence. The revised sentence now reads: “...equipment. Therefore refrigerated...”
Line 85: Etc.? Do not use "etc." as it is necessary to mention each of the ingredients so that the research can be reproducible.
Response: Thanks for your suggestions. We have revised the manuscript to explicitly list all the ingredients without using "etc."
Line 89: Brined? With which type of salt?
Response: Thanks for your suggestions. The beef was marinated for 1 hour with a spice pack, and more details can be found in line 129-133 of the revised manuscript.
Lines 104-113: This section is normally placed before the statistical analysis. The sensory analysis lacks information on how it was carried out and whether water and biscuits were supplied between samples.
Response: We sincerely appreciate your efforts to enhance the clarity and rigor of our manuscript. 1. After careful consideration, we respectfully propose to retain the current place of the sensory evaluation for the following reasons: Sensory evaluation experiments were conducted on the same day at each sampling point and were important in determining the duration of beef storage, with the remaining metrics being measured after the storage period ended. In addition, we have added the consistency of the other metrics with the sensory scores in line 309-311 of the revised manuscript and given the recommended shelf life. 2. We have included a detailed description of the sensory analysis methodology in the revised manuscript. Additionally, we clarified that water was provided to the participants between samples, ensuring that the evaluations were not influenced by previous samples. More details can be found in line 158-164 of the revised manuscript.
Lines 111-112: Were the samples served at room temperature? Give more detail on how the sensory analysis was carried out.
Response: We sincerely appreciate your efforts to enhance the clarity and rigor of our manuscript. Yes, the samples were served at room temperature to ensure that the sensory attributes could be accurately assessed by the panelists. More details can be found in line 158-164 of the revised manuscript.
Lines 115: Provide more details on how the TAC analysis was carried out.
Response: Thanks for your suggestions. TAC was determined following the Chinese standard GB/T 4789.2-2016. Samples were homogenized for 30 s by adding 10-fold sterile NaCl solution, followed by serial dilution (1:10, NaCl solution). Samples of the appropriate dilution (0.1 mL) were selected and spread on plate count agar and incubated for 72 ± 2 hours at 30 ± 1 °C. TAC was expressed as log CFU/g.
Line 122: Provide the full name and in parentheses give the acronym.
Response: Thanks for your suggestions. We have added an introduction to biogenic amines (BAs) with full name and acronym in line 42 of the revised manuscript.
Line 126: Beef or sauced beef?
Response: Thanks for your suggestions. It's sauced beef, we fixed the error in the manuscript.
Line 127: Repeat? Please describe the methodology in the past tense.
Response: Thanks for your suggestions. We have revised the methodology section in the manuscript.
Line 196-197: The authors state: “From Figure 3(b), the pH values of the three groups showed a slight initial decrease followed by a significant increase over time (P < 0.05).” I do not agree, as it can be observed that T1 group remained constant between days 1 to 9.
Response: We appreciate your careful observation. Groups T1, T2 and T3 in the original manuscript have now been changed to control, T1 and T2. Upon reviewing the data presented in Figure 3(b), we have clarified the text to accurately reflect the behavior of the control group. We have amended the statement to indicate that the control group showed stability in pH values between days 1 to 9, while the other groups exhibited the described trends. This correction has been made in the revised manuscript to ensure clarity and accuracy in our findings.
Line 200: This information cannot be understood. To which group does the value 6.4 belong?
Response: We appreciate your careful observation. We modified the mean value of 6.4 to 6.35, 6.41 and 6.37 on day 1 in control, T1, and T2 groups, respectively.
Lines 208-223: Nowhere is it mentioned whether the results obtained consider the beef to be rancid. Please research the oxidation limit that is considered for beef.
Response: We appreciate your comment regarding the oxidation limit for beef. In the revised manuscript, we have included a detailed discussion on the oxidation limits that define rancidity in beef, specifically referencing the relevant literature. TBARS value ≥ 2.0 mg MDA/kg indicates that the meat is not only significantly rancid but also deemed unfit for consumption.
Lines 302- 305: Interesting information.
Response: Thank you for your positive feedback regarding the information presented in lines 302-305. We appreciate your acknowledgment of its relevance and interest in the context of our study.
Furthermore, it is essential that the manuscript be reviewed by a native speaker of English.
Comments on the Quality of English Language
The English could be improved to more clearly express the research.
Response: Thanks for your suggestions. We have revised the manuscript to improve the quality of the English language, ensuring that the research is expressed more clearly and effectively.
Reviewer 2 Report
Comments and Suggestions for Authors All comments are shown in the attached file.When citing references in parentheses, ensure there is a space between
the preceding text and the citation.

Minor errors were noticed and corrected in the attached file.
Author Response
L15: lower comparet to T1? L16: Compared to which group?
Response: Thanks for your suggestions. We have revised the abstract to clarify that tray packing serves as the control treatment in our study. Groups T1, T2 and T3 in the original manuscript have now been changed to control, T1 and T2. The performance of the T1 and T2 groups is compared to the control group.
L 28: Please provide some updated data regarding China’s annual beef production.
Response: Thanks for your suggestions. In the revised manuscript we have added the latest data regarding beef consumption in China. The USDA's Global Beef Market Forecast 2024-2025 report estimates that China's beef consumption will reach 11.557 million tons by 2025.
L 86: add space
Response: Thanks for your suggestions. We have modified it in the revised manuscript, amending it to Xinfadi market in Fengtai District, Beijing, China.
L 88: Brined is the same as marinating for 1h as shown in Figure1? Is it done by heating, since fire is depicted in the figure?
Response: We appreciate your observation. Your suspicion is correct, and we have rewritten this section in L 129-136 of the revised manuscript.
L 102: Please change the title of figure. For example: Processing, packing, and storage of sauced beef samples.
Response: We have changed the title of the figure to "Processing, packing, and storage of sauced beef samples" as suggested. Thank you for your valuable input.
L 127: Are you sure the samples were centrifuged for only 12 seconds? This seems too short. Please check the literature, as a minimum of 5 minutes is typically required.
Response: We appreciate your concern regarding the centrifugation time. Upon reviewing our methodology, we have clarified that the samples were homogenized at 12000 rpm for 12 s, followed by centrifugation at 8,000 g for 5 min.
L 191: Uniform in the text, use abbreviations or the full name for the day(s)
Response: Thanks for your suggestions. We have reviewed the entire manuscript and ensured consistency in the use of full name for the days.
L 251: Use the same font for letters (a), (b), (c) and (d) and the rest of the text.
Response: Thanks for your suggestions. We have revised the manuscript to uniformly use the new Roman font to ensure that the fonts of letters (a), (b), (c) and (d) are consistent with the rest of the text throughout the document.
L 423: Please check the instructions for references, the use of italic font, page range...
Response: Thanks for your suggestions. We have thoroughly reviewed the manuscript to ensure compliance with the reference formatting guidelines. All references have been updated to meet the specified style, including the correct use of italic fonts where required. Additionally, we have verified and corrected the page ranges for all cited works to align with the journal's requirements.
We sincerely appreciate your meticulous attention to grammatical and lexical refinements in our manuscript. Your expertise has significantly elevated the clarity and professionalism of the text, and all suggested corrections have been carefully incorporated into the revised version. Thank you again for your invaluable contributions, which have been instrumental in strengthening the manuscript. We wish you continued success in your scholarly pursuits.
Reviewer 3 Report
Comments and Suggestions for Authors
The article deals with the study of the preservation of packaged meat under different conditions with emphasis on microbiological, sensory and physicochemical aspects.
The paper is interesting and well structured, however, there is room for improvement according to the following:
In the abstract if tray packing is the control should it not be mentioned as a treatment?
In the abstract the concept of imp and hx, tac is still not defined, it would be convenient to define it at least once and then use the abbreviation.
Revise some keywords for better visibility of the work.
In l33 avoid words like love.
l39-41 improve the wording, the way it is, it is too obvious.
in l43-49 It would be convenient to mention some other non-thermal conservation method such as the use of high hydrostatic pressures.
In l55 develop the idea a little more, what is the risk or consequence of the development of biogenic amines?
When talking about conservation methods, it would be useful to add some more qualitative information on the reductions and improvements of these methods.
In l79 avoid the use of words like etc.
The name of treatment for tray packaging needs to be revised.
l85 avoid the use of etc.
Are there any quality specifications for the beef tenderloin used?
l89 temperature and time of blanching, washing and also brining processes?
For all materials used it is necessary to specify properties such as thickness, oxygen and water vapour transfer rate. It is also necessary to specify the vacuum applied in terms of pressure.
3.1 could also include some quantitative information to improve the discussion, as it is very qualitative.
Add error bars to the graphs in figure 2.....
Revise the scale of figure 3a, it should only go up to 13.
It would be convenient to standardise the colours and patterns used for each treatment. In some graphs there are different colours and this can lead to misinterpretation.
Eliminate 5a as it is not a result.
l302-319 contains rather theoretical information that could be in the introduction section.
Rewrite the conclusions with emphasis on the findings and knowledge generated, also reinforce at the end of the conclusions the practical application.
Author Response
In the abstract if tray packing is the control should it not be mentioned as a treatment?
Response: Thanks for your suggestions. We have revised it in the abstract.
In the abstract the concept of imp and hx, tac is still not defined, it would be convenient to define it at least once and then use the abbreviation.
Response: Thanks for your suggestions. We have provided the full name for these concepts in the abstract.
Revise some keywords for better visibility of the work.
Response: Thanks for your suggestions. We have revised the keywords to enhance the visibility of our work.
In l33 avoid words like love.
Response: Thanks for your suggestions. We have revised line 33 to remove the word 'love' and replaced it with a more neutral term to maintain the academic tone of the paper.
l39-41 improve the wording, the way it is, it is too obvious.
Response: Thanks for your suggestions. We have revised this sentence in the manuscript.
in l43-49 It would be convenient to mention some other non-thermal conservation method such as the use of high hydrostatic pressures.
Response: We appreciate this suggestion and have included a discussion of high hydrostatic pressures and cold atmospheric plasma as non-thermal conservation methods in the revised manuscript.
In l55 develop the idea a little more, what is the risk or consequence of the development of biogenic amines?
Response: Thanks for your suggestions. We have added a description of the hazards of biogenic amines in the introduction section, this addition can be found in the revised section (L44-49). Biogenic amines (BAs) are nitrogenous compounds beneficial in moderation, but excessive intake (e.g., histamine > 40 mg) causes headaches, vomiting, blood pressure fluctuations, or even fatal toxicity.
When talking about conservation methods, it would be useful to add some more qualitative information on the reductions and improvements of these methods.
Response: Thanks for your suggestions. We have added a section in the manuscript that provides qualitative information regarding the reductions and improvements achieved through the various conservation methods introduced.
In l79 avoid the use of words like etc.
Response: Thanks for your suggestions. We have carefully revised and corrected these errors in the manuscript.
The name of treatment for tray packaging needs to be revised.
Response: Thanks for your suggestions. We have revised the name to clarify that tray packing serves as the control group in our study.
l85 avoid the use of etc.
Response: Thanks for your suggestions. We have carefully revised and corrected these errors in the manuscript.
Are there any quality specifications for the beef tenderloin used?
Response: Thanks for your suggestions. Tenderloins from cattle with an average age of 1.5 to 2 years were used as raw meat.
l89 temperature and time of blanching, washing and also brining processes?
Response: Thanks for your suggestions. The cubes were blanched for 5 min and washed to remove foam, then the beef was marinated for 1 h with a spice pack, the time was measured from the boiling of the water, and more details can be found in line 126-130 of the revised manuscript.
For all materials used it is necessary to specify properties such as thickness, oxygen and water vapour transfer rate. It is also necessary to specify the vacuum applied in terms of pressure.
Response: Thanks for your suggestions. We have now included detailed specifications regarding the properties of polyethylene (PE) cling film and vacuum-packed bags. The PE cling film has a thickness of 25 μm, an O2 transmission rate (OTR) of 4000 cm³/ (m²·day·atm), and a water vapor transmission rate (WVTR) of 15 g/m²·day. In comparison, the PE vacuum bag has a thickness of 80 μm, an OTR of 50 cm³/ (m²·day·atm), a WVTR of 2 g/m²·day, and a vacuum pressure of approximately 0.77 atm.
3.1 could also include some quantitative information to improve the discussion, as it is very qualitative.
Response: Thanks for your suggestions. We have revised section 3.1 to include quantitative data that supports our qualitative findings.
Add error bars to the graphs in figure 2.....
Response: We sincerely appreciate your efforts to enhance the clarity and rigor of our manuscript. After careful consideration, we respectfully propose to retain the current design of the radar chart without error bars for the following reasons: as radar charts present standardized sensory dimensions, error bars may compromise visual clarity.
Revise the scale of figure 3a, it should only go up to 13.
Response: Thanks for your suggestions. We have carefully revised and corrected this error in the manuscript.
It would be convenient to standardise the colours and patterns used for each treatment. In some graphs there are different colours and this can lead to misinterpretation.
Response: Thanks for your suggestions. We have standardized the colours and patterns used for each treatment across all graphs in the revised manuscript.
Eliminate 5a as it is not a result.
Response: Thanks for your suggestions. We have removed figure 5(a) from the manuscript as per your suggestion.
l302-319 contains rather theoretical information that could be in the introduction section.
Response: Thanks for your suggestions. We have revised the manuscript and moved the theoretical information from line 302-319 to line 50-66.
Rewrite the conclusions with emphasis on the findings and knowledge generated, also reinforce at the end of the conclusions the practical application.
Response: Thanks for your suggestions. We have revised the conclusions to better highlight the key findings of our research.
Round 2
Reviewer 3 Report
Comments and Suggestions for Authors
The authors responded satisfactorily to most of my observations and comments. The clarity and quality of the article improved considerably.